# Aberrant methylation and expression of TNXB promote chondrocyte apoptosis and extracullar matrix degradation in hemophilic arthropathy via AKT signaling

Jiali Chen[1,2†], Qinghe Zeng[1,2†], Xu Wang[1,2], Rui Xu[3], Weidong Wang[4], Yuliang Huang[4], Qi Sun[5], Wenhua Yuan[1], Pinger Wang[1], Di Chen[6,7]*, Peijian Tong[8]*, Hongting Jin[1]*

[1]Institute of Orthopaedics and Traumatology, the First Affiliated Hospital of Zhejiang Chinese Medical University, Zhejiang Provincial Hospital of Chinese Medicine, Hangzhou, China; [2]The First College of Clinical Medicine, Zhejiang Chinese Medical University, Hangzhou, China; [3]Department of Orthopaedics, Affiliated Hospital of Jiangxi University of Traditional Chinese Medicine, Nanchang, China; [4]Department of Osteology, The Second Affiliated Hospital of Zhejiang Chinese Medical University, Hangzhou, China; [5]Department of Orthopaedic Surgery, Fuyang Orthopaedics and Traumatology Affiliated Hospital of Zhejiang Chinese Medical University, Hangzhou, China; [6]Research Center for Computer-aided Drug Discovery, Shenzhen Institute of Advanced Technology, Chinese Academy of Sciences, Shenzhen, China; [7]Faculty of Pharmaceutical Sciences, Shenzhen Institute of Advanced Technology, Chinese Academy of Sciences, Shenzhen, China; [8]Department of Orthopaedic Surgery, the First Affiliated Hospital of Zhejiang Chinese Medical University, Zhejiang Provincial Hospital of Chinese Medicine, Hangzhou, China

*For correspondence:
di.chen@siat.ac.cn (DC);
tongpeijian@163.com (PT);
hongtingjin@163.com (HJ)

†These authors contributed equally to this work

**Abstract** Recurrent joint bleeding in hemophilia patients frequently causes hemophilic arthropathy (HA). Drastic degradation of cartilage is a major characteristic of HA, but its pathological mechanisms has not yet been clarified. In HA cartilages, we found server matrix degradation and increased expression of DNA methyltransferase proteins. We thus performed genome-wide DNA methylation analysis on human HA (N=5) and osteoarthritis (OA) (N=5) articular cartilages, and identified 1228 differentially methylated regions (DMRs) associated with HA. Functional enrichment analyses revealed the association between DMR genes (DMGs) and extracellular matrix (ECM) organization. Among these DMGs, Tenascin XB (TNXB) expression was down-regulated in human and mouse HA cartilages. The loss of *Tnxb* in *F8*[−/−] mouse cartilage provided a disease-promoting role in HA by augmenting cartilage degeneration and subchondral bone loss. *Tnxb* knockdown also promoted chondrocyte apoptosis and inhibited phosphorylation of AKT. Importantly, AKT agonist showed chondroprotective effects following *Tnxb* knockdown. Together, our findings indicate that exposure of cartilage to blood leads to alterations in DNA methylation, which is functionally related to ECM homeostasis, and further demonstrate a critical role of TNXB in HA cartilage degeneration by activating AKT signaling. These mechanistic insights allow development of potentially new strategies for HA cartilage protection.

## eLife assessment

This **important** study identifies the TNXB-AKT pathway as a potential mechanism underlying hemophilia-associated cartilage degeneration. The evidence supporting the conclusions is **convincing**, with murine and human patient evidence as well as genome-wide DNA methylation analysis. This paper would be of interest to cell biologists and biochemists working on the field of musculoskeletal disorders.

## Introduction

Hemophilia is a X-linked bleeding disorder due to the deficiency of coagulation factor VIII (hemophilia A) and IX (hemophilia B). As a one of the most frequent rare diseases, hemophilia is a persistent, challenging condition for patients, physicians, and society (*Carulli et al., 2021*). Joint bleeding (hemarthrosis), especially in knee joint, is the most common clinical manifestation in patients with severe hemophilia (i.e. plasma FVIII or FIX <1 U/dl; *Gualtierotti et al., 2021*). Recent evidence also indicates that hemarthrosis may occur spontaneously in patients with moderate (plasma factor levels of 1–5 UI/dl) or mild disease (plasma factor levels of >5 UI/dl; *Di Minno et al., 2013*; *Wyseure et al., 2019*). Repeated episodes of hemarthrosis eventually result in hemophilic arthropathy (HA), a debilitating and irreversible condition. HA often starts at an early age and characterized by joint impairment, chronic pain, and reduced quality of life (*Aledort et al., 1994*; *Bolton-Maggs and Pasi, 2003*; *Fischer et al., 2005*; *Manco-Johnson et al., 2007*; *Oldenburg, 2015*; *Olivieri et al., 2012*).

Current efforts to prevent the development of HA are mainly focused on management of acute joint bleeding and optimizing prophylactic replacement therapy (*Gualtierotti et al., 2021*; *Jackson et al., 2015*; *Schramm et al., 2012*). However, the widespread adoption of the modern hematological care has not conferred considerable protection against the development of HA; even in patients with hemophilia receiving intermediate- and high-dose prophylaxis, many still develop HA (*Kleiboer et al., 2022*). Pathological changes of HA begin with the first joint bleeding, either breakthrough or sub-clinical, usually during the first years of life (*Wyseure et al., 2019*; *Wojdasiewicz et al., 2018*). They consist in progressive and irreversible alterations induced by the direct and indirect toxicity of the free blood in the joint space. The joint replacement surgery, such as Total Knee Arthroplasty (TKA), is considered an intervention for alleviating pain and restoring joint function in patients with end-stage HA. However, it is associated with a high incidence of complications, primarily attributed to septic loosening and recurrent postoperative bleeding (*Kleiboer et al., 2022*). Given no disease modifying therapy available to intervene in the HA perpetuating process, HA remains a persistent problem and challenge for hemophilia patients. Thus, to understand the pathogenesis of HA and subsequently explore possible targets for therapy is necessary.

One distinctive pathophysiological manifestation of HA is cartilage degeneration (*Pulles et al., 2017*). Cartilage is a rather inert tissue, consisting of matrix proteins and only one cell type, chondrocytes that are responsible for production of matrix synthesis and rely on synovial fluid for nutrients as cartilage lacks blood supply (*Fermor et al., 2007*; *Lafont, 2010*). Nevertheless, recurrent intra-articular bleeding creates a 'toxic' environment to cartilage, by inducing synovial inflammation, hemosiderin accumulation and excessive mechanical stress (*Pulles et al., 2017*; *van Vulpen et al., 2018*). As a consequence, dysregulation of metabolism and abnormal apoptosis occur in chondrocytes, ultimately resulting in deterioration of the cartilage matrix (*Gualtierotti et al., 2021*).

DNA methylation, a core epigenetic mechanism, is high dynamic and susceptible to cues from the environment (*Feil and Fraga, 2012*; *Jirtle and Skinner, 2007*). DNA methylation is the addition of a methyl group to the cytosine residue of DNA molecules, which is catalyzed by DNA methyl transferase (Dnmt) family of proteins that is comprised of three members: DNMT1, DNMT3A, and DNMT3B. DNA methylation is mainly located in the gene regulatory region, usually repressing gene expression by blocking the transcriptional accessibility of regulatory genomic regions (*Stelzer et al., 2015*). Recently, multiple studies confirmed the important role of abnormal DNA methylation in the pathogenesis of arthritis (*Wang et al., 2017*; *Zhang et al., 2016*). However, the epigenetic mechanisms underlying HA-related cartilage degradation have not been explored.

In this study, we conducted a genome-wide DNA methylation study with the goal of identifying differentially methylated genes (DMGs) and pathways for HA. Further, we knocked down *Tnxb*, a key DMG, in primary mouse chondrocytes and hemophilia A (*F8*$^{-/-}$) mice. The biological function and underlying mechanisms of TNXB in HA progression were also investigated *in vivo* and *in vitro*. Our

findings provide a new mechanism for articular cartilage lesion in HA, which may lead to the discovery of novel therapeutic targets for the treatment of HA.

## Results

### Articular cartilages of HA patients show severe damage and aberrant elevations of DNMT1 and DNMT3A proteins

In this study, we collected cartilage samples in tibial plateau from OA and HA patients undergo total knee replacement (TKR) surgery. Consistent with previous reports, the HA cartilage exhibited severe damage and significant hemosiderin deposition, compared with that of OA (*Figure 1A*). MRI analysis revealed the subchondral bone loss in HA patients (*Figure 1B*). ABH staining further assessed the cartilage degeneration, and detected a substantial sulfated glycosaminoglycan (sGAG) depletion in HA cartilages (*Figure 1C, D*). As expected, comparing to OA, HA cartilages showed prominent decrease in cartilage matrix protein COL2A1 and increase in expression of MMP13, the primary enzymes responsible for cartilage degeneration (*Figure 1E, F*). Additionally, obvious increase in DNMT1 and DNMT3A protein levels was also detected in HA cartilages (*Figure 1G, H*), whereas no significant changes were found in DNMT3B (*Figure 1-Figure supplement 1A, B*). These data confirm the severe damage in articular cartilage and subchondral bone of HA patients.

### Genome-wide DNA methylation analysis reveals the epigenetic landscape of HA cartilage

To understand the methylation profile associated with HA cartilage degeneration, genome-wide DNA methylation alterations were examined in 10 subjects (5 OA cartilages and 5 HA cartilages) by using 850 K ChIP. The methylation levels of whole genome-wide CpGs were in a classic bimodal distribution where most CpGs were either slightly or highly methylated (*Figure 2A*). Of note, global DNA methylation levels exhibited systematic differences between HA and OA, as illustrated by their separation pattern in principle component analysis (*Figure 2B*). Overall, 1228 significant differentially methylated regions (DMRs; $p < 0.05$ after the Benjamini & Hochberg correction for multiple testing) were identified, with a mean fold change in methylation difference of 0.90 (*Figure 2C*, and *Supplementary file 3a*). Compared with the OA, 69.95% and 30.05% DMRs were hypermethylated and hypomethylated in HA, respectively (*Figure 2—figure supplement 1*). Given the specificity of DMR, we then determined whether the identified DMRs were preferentially colocalized with some specific genomic features. The large majority (93.49%) of these DMRs were located in gene promoters, 5'-UTRs, 3'-UTRs, and exons, with only 6.51% overlapping intergenic regions (*Figure 2—figure supplement 2*).

To determine the biological functions of these significant DMRs, GO and KEGG enrichment analysis were performed on DMR-related genes (DMGs). GO analysis revealed enrichment for terms largely related to extracellular matrix (ECM) organization, skeletal system development (*Figure 2D*, *Supplementary file 3b*). The top two KEGG pathways were ECM-receptor interaction ($p$-value = 5.2 × 10$^{-6}$) and PI3K-Akt signaling pathway ($p$-value = 5.3 × 10$^{-4}$; *Figure 2E*, *Supplementary file 3c*). DMGs presented in these pathways have mostly been proved to play an important role in cartilage matrix homeostasis, such as *COL1A1*, *COL1A2*, *COMP Chen et al., 2019*; *Maly et al., 2021*; *Zhang et al., 2019*. Among them, we found TNXB were significantly low expressed in human HA cartilage compared to OA cartilage by using IHC staining (*Figure 2F, G*). However, TNXB was not differentially expressed in OA and HA synovial membranes (*Figure 2—figure supplement 3A, B*).

### Decreased expression of Tnxb is associated with cartilage degradation by joint bleeding in *F8*$^{-/-}$ mice

To assess the potential relevance of TNXB in HA *in vivo*, we established a mouse model for HA, in which *F8*$^{-/-}$ mice received a needle puncture injury to the knee joint *Haxaire et al., 2018*; *Magisetty et al., 2022*. This needle puncture induced excessive bleeding and caused severe hemarthrosis (*Figure 3—figure supplement 1A, B*). Analysis of cartilage tissue sections by Toluidine blue staining uncovered that joint bleeding remarkably promoted cartilage degeneration at 4 and 8 weeks after injury (*Figure 3A, B*). IHC staining further demonstrated lower expression of Col2a1 and increased expression of Mmp13 (*Figure 3C-F*). Next, micro-CT was used to assess the influence of hemarthrosis in osteogenesis. An obvious osteopenia was detected in subchondral bone at 4 and 8 weeks post

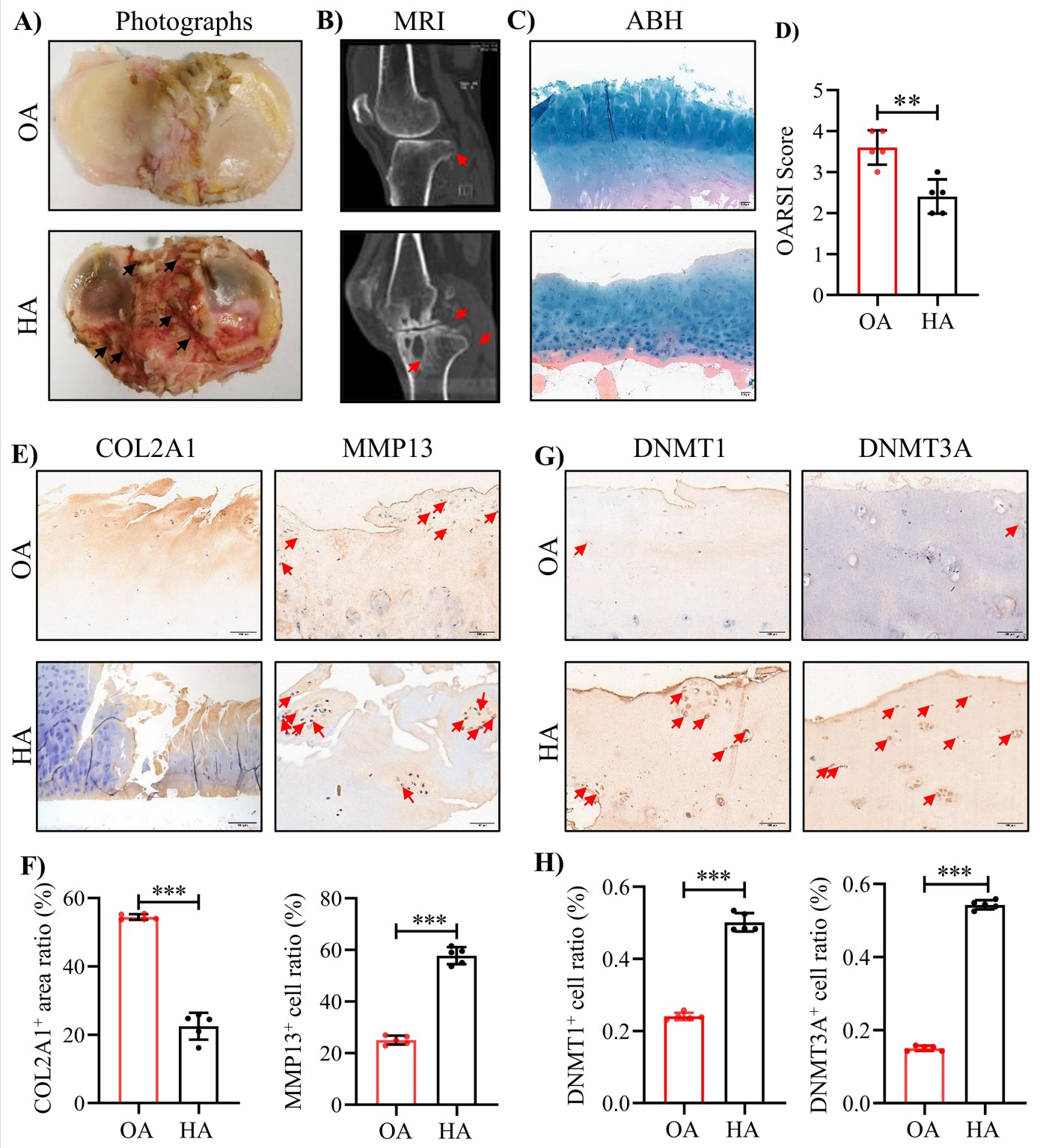

**Figure 1.** The severe cartilage damage in human HA. (**A**) Human cartilage samples in the tibial plateau were obtained from patients with OA and HA. Black arrow indicates hemorrhagic ferruginous deposits. (**B**) MRI imaging of the knee joint in patients with HA and OA. Red arrow indicates the wear area. (**C**) Representative images of ABH/OG staining for HA and OA cartilage. (**D**) The degree of cartilage degeneration was quantified according to the OARSI score. (**E**) Representative IHC staining of COL2A1 and MMP13. (**F**) Quantification of the proportion of COL2A1 and MMP13 positive regions in human cartilage. (**G**) Representative IHC staining of DNMT1 and DNMT3A. (**H**) Quantification of the proportion of DNMT1 and DNMT3A positive cells

*Figure 1 continued on next page*

*Figure 1 continued*

in human cartilage. Red arrows indicate positive cells. Scale bar: 100 µm. Data were presented as means ± SD; n = 5 per group. And analyzed by two-tailed unpaired parametric Student's t test, **p < 0.01, ***p < 0.001.

The online version of this article includes the following figure supplement(s) for figure 1:

**Figure supplement 1.** Representative immunofluorescence images (**A**) and quantification (**B**) of DNMT3B in human HA and OA cartilages.

injury (*Figure 3—figure supplement 2A-D*). As expected, the above histopathological changes in F8$^{-/-}$ mice following needle puncture were similar with what we observed in Human HA. Meanwhile, decreased expression of Tnxb and increased expressions of Dnmt1 and Dnmt3a were showed in articular cartilages 4 and 8 weeks following injury (*Figure 3G-L*). Above results suggest that the suppression of TNXB in cartilage is associated with HA development.

## TNXB knockdown impairs chondrocyte metabolism and aggravates the progression of HA

To verify the relationship between TNXB and methylation levels, we treated primary mouse chondrocytes with DNA Methyltransferase inhibitors RG108 or 5-Aza-dc, well-known to block DNA methylation (*He et al., 2022*; *Zhu et al., 2019*). After 24 hr treatment, RG108 or 5-Aza-dc (10, 25 µM) significantly up-regulated the mRNA level of *Tnxb*, indicating that the DNA methylation inhibits the expression of TNXB in chondrocytes (*Figure 4—figure supplement 1A, B*). To further identify the biological functions of TNXB in HA, we first knocked it down by specific siRNA in primary mouse chondrocytes. The efficiency of *Tnxb* knockdown was confirmed by results of qPCR and Western blot experiments (*Figure 4A-C*). When compared with the siRNA-control group, we found that *Tnxb* knockdown led to a decrease in *Col2a1* mRNA expression and an increase in *Mmp13* mRNA expression (*Figure 4D, E*). Western blot and immunofluorescence staining analysis also showed a downregulated expression of Col2a1 protein and upregulated expression of Mmp13 protein (*Figure 4E-H*). Those observations reveals that TNXB positively regulates the metabolism of chondrocyte ECM.

Subsequently, we explored the effects of TNXB on HA cartilage degradation *in vivo*, using intra-articular injection of adeno-associated virus (AAV) carrying *Tnxb*-specific short hairpin RNA (shRNA) in F8$^{-/-}$ mice. Four weeks after injection, more severe cartilage degeneration was observed in *Tnxb*-KD mice, compared with vehicle group (*Figure 4I, J*). In addition, *Tnxb*-KD mice showed decreased expression of Col2a1 and increased expression of Mmp13 in cartilages (*Figure 4K, L*). The subchondral bone of *Tnxb*-KD mice was significantly less than vehicle group mice (*Figure 4—figure supplement 2A-E*). Collectively, the above results imply that *TNXB* knockdown in cartilage accelerated the HA development.

## TNXB deficiency sensitizes chondrocytes to apoptosis by inhibiting AKT activation

Since blood exposure in joint cavity strongly causes chondrocyte apoptosis (*Figure 5—figure supplement 1A, B*), we next explored the functional mechanisms of TNXB in HA chondrocyte apoptosis. *Tnxb* knockdown induced the apoptosis in chondrocytes, as evidenced by an increased proportion of Tunel-positive cells (*Figure 5A, B*). Meanwhile, protein expression levels of pro-apoptotic Bax and Cleaved-Caspase3 were significantly increased in *Tnxb*-KD chondrocytes, while anti-apoptotic Bcl-2 was decreased (*Figure 5C-F*). Moreover, AAV-shTnxb injection also increased the number of Tunel-positive cells in articular cartilages in HA mice (*Figure 5G, H*). IHC staining confirmed that Bcl-2 expression was significantly reduced in *Tnxb*-KD HA mice accompanied by enhanced expressions of Bax and Cleaved-Caspase-3 (*Figure 5I, L*).

It is well known that the PI3K/AKT signaling pathway regulates apoptosis (*Cai et al., 2019*; *Hu et al., 2018*; *Luo et al., 2003*), and our KEGG analysis and IHC staining confirmed the involvement of this pathway in HA cartilage degeneration (*Figure 5M, N*). We, therefore, analyzed PI3K/AKT pathway following *Tnxb* knockdown. In *Tnxb*-KD chondrocytes, p-AKT1 protein level decreased significantly, while AKT1 level remained unchange (*Figure 6A, D*). Next, we sought to investigate whether AKT1 activation could attenuate chondrocyte apoptosis induced by *Tnxb* knockdown. AKT1 agonist SC79 was used to treated *Tnxb*-KD chondrocytes, and then its efficiency was verified by western blotting (*Figure 6—figure supplement 1A, B*). Twenty-four hr after SC79 treatment, the expressions of Bcl-2

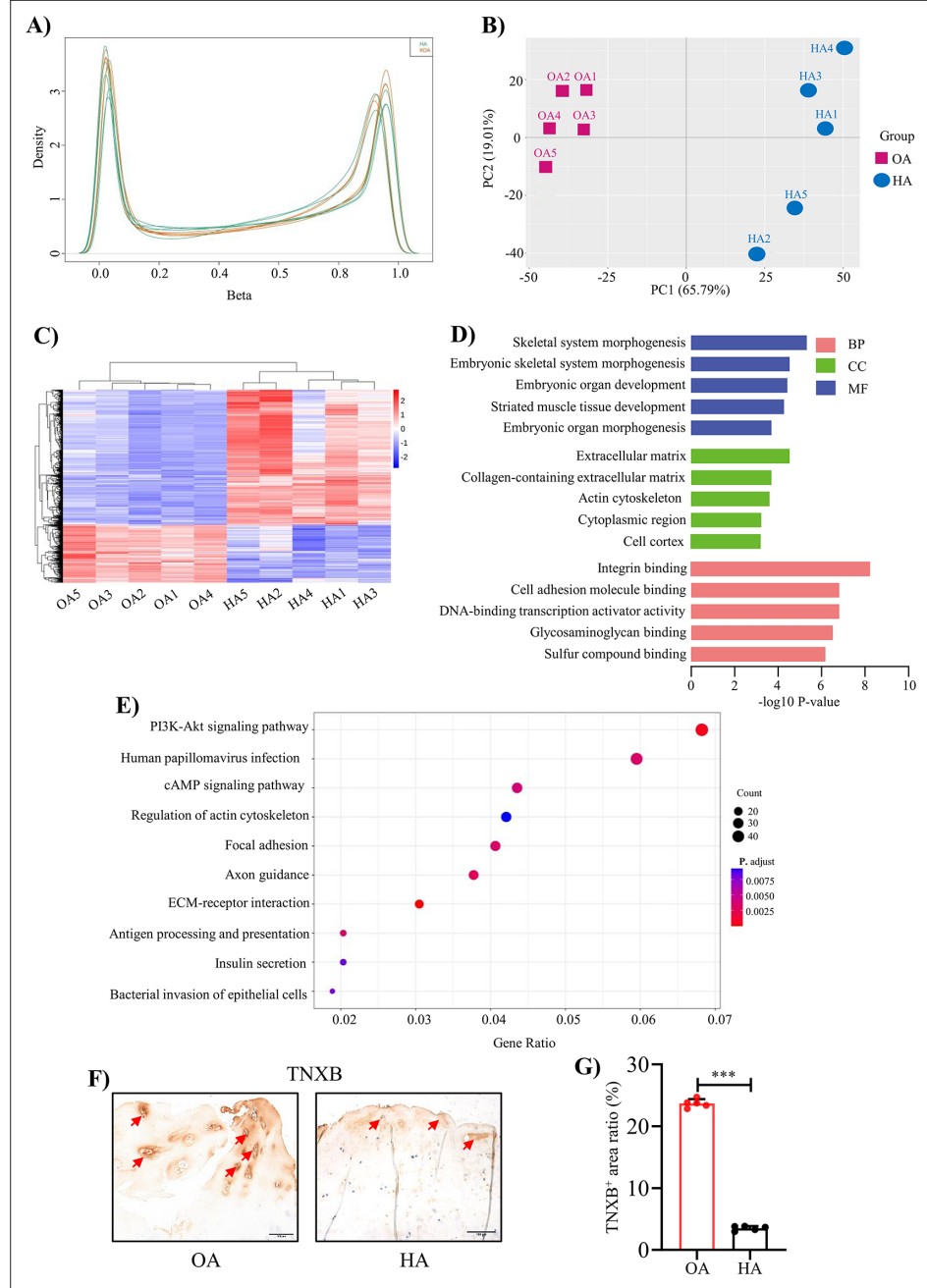

**Figure 2.** Genome-wide DNA methylation profile in HA cartilage and biological functions of DMGs. (**A**) Distribution of methylation level density of CpGs. Note: X = degree of methylation; Y = the CpG site density corresponding to the level of methylation. (**B**) Principal component analysis of DNA methylation data. (**C**) Heatmap shows the 700 significant DMRs between HA and OA. (**D**) Enriched GO terms for DMR-related genes. (**E**) KEGG enrichment analysis of DMR-related genes. (**F**) Representative IHC staining of TNXB. Red arrows indicate positive areas. Scale bar: 100 μm. (**G**) Quantification of the proportion of TNXB positive regions in human cartilage. Data were presented as means ± SD; n = 5 per group. And analyzed by two-tailed unpaired parametric Student's t test, ***$p < 0.001$.

The online version of this article includes the following figure supplement(s) for figure 2:

**Figure supplement 1.** The proportion of hypermethylated and hypomethylated DMRs.

**Figure supplement 2.** Proportion of DMRs for each genetic feature.

**Figure supplement 3.** Representative IHC staining (**A**) images and (**B**) corresponding quantification of TNXB in synovium from human HA and OA.

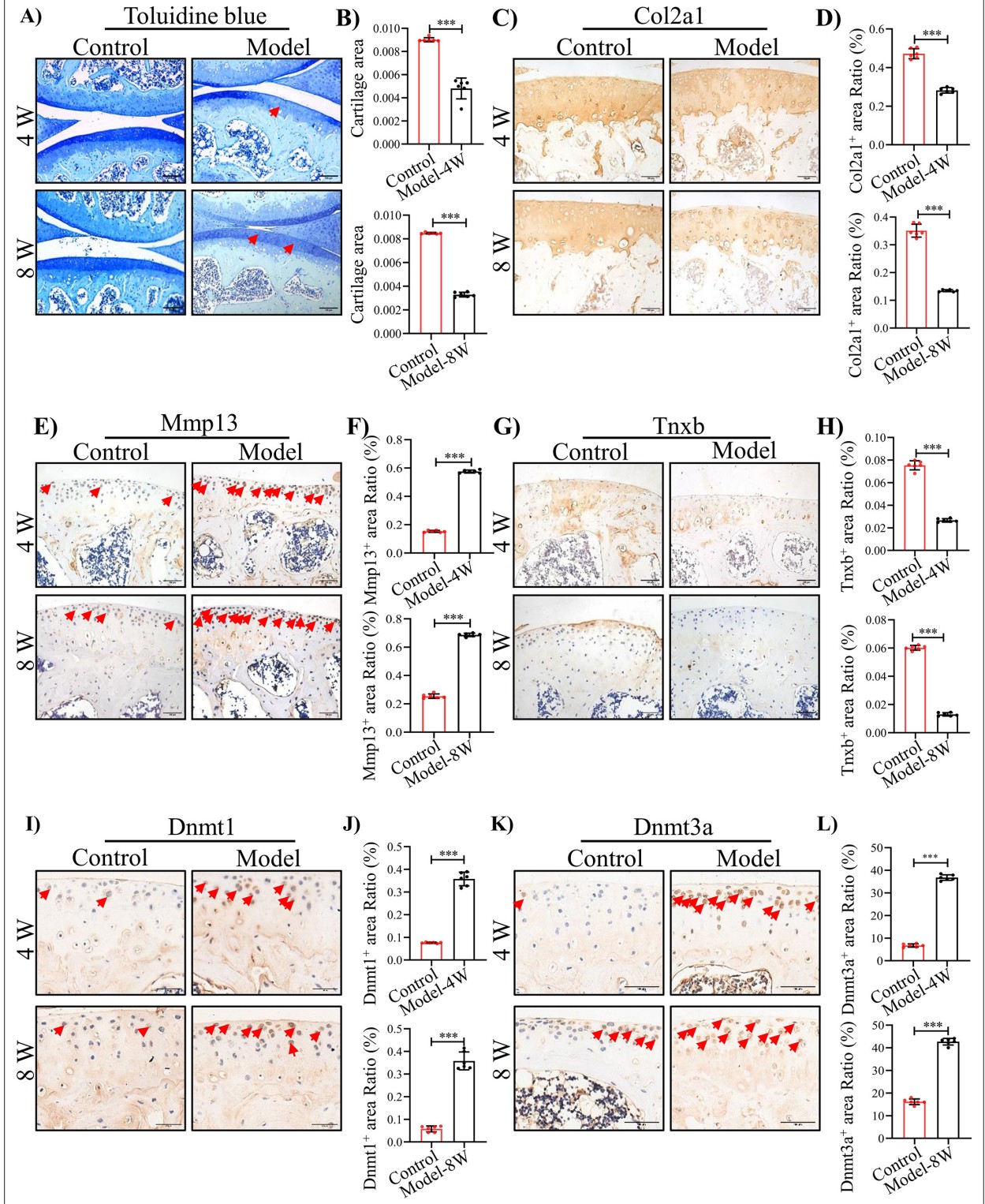

**Figure 3.** TNXB expression is drastically reduced in HA mouse cartilages. (**A**) Representative images and (**B**) Quantitative analysis of Toluidine blue staining for knee sections of *F8$^{-/-}$* mice at 4 and 8 weeks following injury. Representative IHC staining of (**C**) Col2a1, (**E**) Mmp13, (**G**) Tnxb, (**I**) Dnmt1, and (**K**) Dnmt3a in *F8$^{-/-}$* mice at 4 and 8 weeks post injury. Quantification of the proportion of (**D**) Col2a1, (**F**) Mmp13, (**H**) Tnxb, (**J**) Dnmt1 and (**L**) Dnmt3a positive regions. Scale bar: 100 μm. Data were presented as means ± SD; n = 6 mice per group. And analyzed by two-tailed unpaired parametric Student's t test, ***$p < 0.001$.

The online version of this article includes the following figure supplement(s) for figure 3:

*Figure 3 continued on next page*

*Figure 3 continued*

**Figure supplement 1.** Representative images of bleeding knee joint of *F8⁻/⁻* mice at (**A**) 4 and (**B**) 8 weeks following injury.

**Figure supplement 2.** Representative 3D reconstruction of the knee joint and subchondral bone of HA model mice at (**A**) 4 weeks and (**B**) 8 weeks after puncture-induced injury.

and Col2a1 (*Figure 6A, B, F*) were upregulated, while Bax, Mmp13, and Cleaved-Caspase9 expressions were markedly downregulated in *Tnxb*-KD chondrocytes (*Figure 6A, C, E, G*). These findings indicate that *TNXB* knockdown induces chondrocyte apoptosis and ECM degradation through regulating AKT activation.

## Discussion

HA, a common manifestation of hemophilia, is typical characterized by dramatic cartilage degradation (*Bolton-Maggs and Pasi, 2003*). Thus, the goal of our work was to provide a novel insight into pathogenic mechanisms underlying HA cartilage degeneration. *Figure 6H* summarized our findings. Genome-wide DNA methylation analysis revealed the DNA methylation changes in cartilages from hemophilia patients and identified *Tnxb* gene associated with HA. The decreased expression of TNXB protein was also confirmed in both human and mouse HA cartilages. Further, *Tnxb* knockdown promoted ECM catabolism and chondrocyte apoptosis, and eventually accelerated the development of HA in *F8⁻/⁻* mice. Meanwhile, decreased TNXB expression inhibited the activation of AKT1 *in vivo* and *in vitro*. Notably, AKT agonist-SC79 enhanced ECM synthesis and suppressed apoptosis in *Tnxb*-KD chondrocytes. From these findings, it could be hypothesized that abnormal methylation and decreased expression of TNXB are responsible for disruption of cartilage homeostasis in HA.

In hemophlia patients, recurrent joint bleeding creates a toxic environment including multifaceted processes such as hemosiderin deposition, inflammatory response and mechanical pressure overload (*Hooiveld et al., 2003a*; *Srivastava, 2015*; *van Vulpen et al., 2015*; *Visser et al., 2015*). DNA methylation is a dynamic epigenetic modification in response to environment. Since there are fewer epigenetic studies on the pathological mechanisms of HA, we performed genome-wide DNA methylation analysis, we found 1228 significant DMRs associated with HA with 69.95% hypermethylated, which was also supported by a corresponding increase in DNMT1 and DNMT3A protein levels in HA cartilages. These results are similar to previous reports showing that aberrant elevations of Dnmt1 and Dnmt3a promote cartilage degeneration in destabilization of medial meniscus and aging models (*Zhu et al., 2019*; *Iijima et al., 2023*).

DNA methylation at gene promoter region is one of the important epigenetic mechanisms in the regulation of gene expression (*Küçük et al., 2015*; *Kulis and Esteller, 2010*). The identified HA-related DMRs showed high proportion in transcriptional start sites. These data suggest that DNA methylation is a critical mediator in the pathology of HA. Nevertheless, due to the severe erosion of HA cartilage, the mRNA levels of DMR-related genes were not explored in this study. Besides, our sample size for DNA methylation sequencing is relatively small. Thus, further studies using additional more HA samples will be conducted to investigate the possible transcriptional alteration of identified genes as well as their potential roles in HA development.

Furthermore, our enrichment analysis showed that DMR-related genes were significantly enriched in ECM organization and ECM-receptor interaction terms. It is consistent with previous reports that ECM acts as an epigenetic informational entity capable of transducing and integrating intracellular signals via ECM-receptor interaction (*Jones and Jones, 2000*; *Kim et al., 2011*; *Alcaraz et al., 2014*). DMR-related genes enriched in these terms, many are known to be critical compositions of cartilage matrix, such as *COMP*, *COL1A1*, and *COL1A2* (*Chen et al., 2019*; *Maly et al., 2021*; *Zhang et al., 2019*). In Articular cartilage, ECM provides nutrients and mechanical support for chondrocytes (*Sutherland et al., 2015*), and its age-related stiffening epigenetically regulates gene expression and compromises chondrocyte integrity (*Iijima et al., 2023*). Above findings highlight that ECM-related genes are potential candidates for HA.

Tenascin-X (TNXB), an ECM glycoprotein, is the top differentially methylated gene in our DNA methylation analysis of HA cartilages. In addition, DNA methylation of *TNXB* has also been reported in whole blood from rheumatoid arthritis patients and in retinal pigment epithelium from patients with age-related macular degeneration (*Anaparti et al., 2020*; *Porter et al., 2019*). Using IHC staining, we

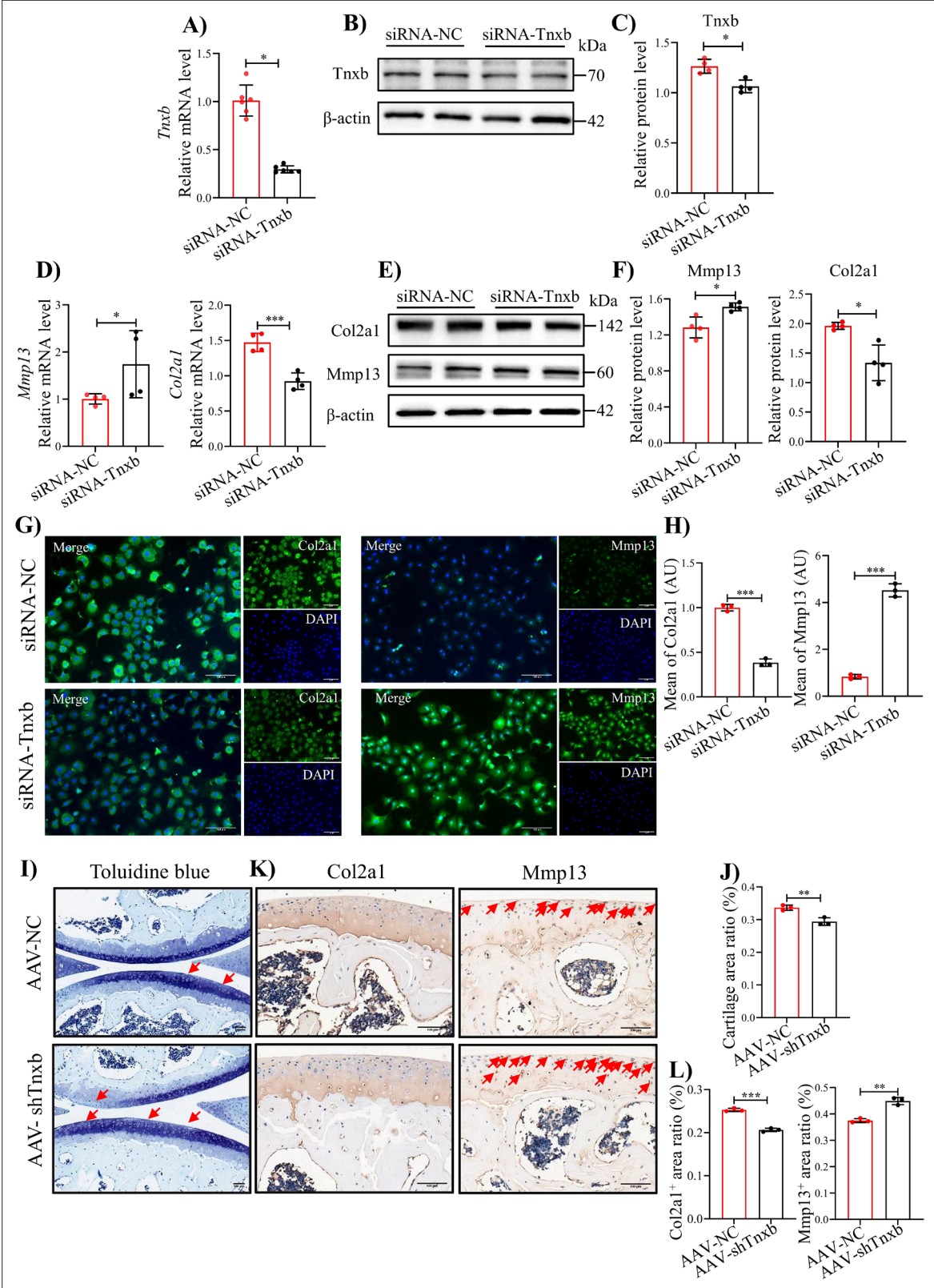

**Figure 4.** knockdown of *Tnxb* in chondrocytes induces extracellular matrix degradation and accelerates the progression of HA. (**A**) qPCR and (**B**) western blotting analysis for Tnxb in mouse chondrocytes treated with siRNA-NC or siRNA-*Tnxb*. (**C**) Corresponding quantification analysis of Tnxb protein. Quantification of mRNA levels for (**D**) *Mmp13* and*Col2a1* in *Tnxb*-KD chondrocytes. (**E**) Western blot and (**F**) corresponding quantification analysis for Mmp13 and Col2a1 protein. (**G**) Representative immunofluorescence images of Col2a1 and Mmp13 expression in *Tnxb*-KD chondrocytes.

*Figure 4 continued on next page*

*Figure 4 continued*

(**H**) Quantification of Col2a1 and Mmp13 fluorescence intensity. Representative images of (**I**) Toluidine blue staining and (**K**) IHC staining of Col2a1 and Mmp13 for knee sections of $F8^{-/-}$ mice at 4 weeks after Intra-articular injection of AAV-sh*Tnxb*. (**J**) Quantitative detection of the area of the tibial cartilage area. (**L**) Quantification of the proportion of Col2a1 and Mmp13 positive regions. Red arrow indicates the wear area. Scale bar: 100 μm. Data were presented as means ± SD; n ≥ 3 in each group. And analyzed by two-tailed unpaired parametric Student's t test, *$p < 0.05$, **$p < 0.01$, ***$p < 0.001$.

The online version of this article includes the following source data and figure supplement(s) for figure 4:

**Source data 1.** Original file for the western blot analysis of β-actin in *Figure 4B*.

**Source data 2.** Original file for the western blot analysis of Tnxb in *Figure 4B*.

**Source data 3.** PDF containing *Figure 4B* and original scans of the relevant western blot analysis (anti-β-actin and anti-Tnxb) with highlighted bands and sample labels.

**Source data 4.** Original file for the western blot analysis of β-actin in *Figure 4E*.

**Source data 5.** Original file for the western blot analysis of Col2a1 in *Figure 4E*.

**Source data 6.** Original file for the western blot analysis of Mmp13 in *Figure 4E*.

**Source data 7.** PDF containing *Figure 4E* and original scans of the relevant western blot analysis (anti-β-actin, anti-Col2a1, and anti-Mmp13) with highlighted bands and sample labels.

**Figure supplement 1.** The qPCR analysis for the mRNA level of *Tnxb* in mouse chondrocytes treated with (**A**) RG-108 or (**B**) 5-Aza-dc.

**Figure supplement 2.** The micro-CT analysis of subchondral bone in *Tnxb*-KD HA mice.

detected the downregulated expression of TNXB in human and mouse HA cartilages. TNXB belongs to tenascin family, whose members (TNC, TNR, TNXB, and TNW) share a similar domain pattern: an N-terminal oligomerization domain, a series of epidermal growth factor-like repeats, a variable number of fibronectin-type III (FNIII) repeats and a C-terminal fibrinogen-like domain (*Tucker et al., 2006*). Emerging evidence suggests the involvement of TNC in cartilage development and degeneration in arthritis (*Hasegawa et al., 2018*; *Hasegawa et al., 2020*). Here, we prove for the first time a role of TNXB in cartilage homeostasis *in vivo* and *in vitro*. Specifically, *Tnxb* knockdown contributed to an increase in Mmp13 expression and decrease in Col2a1 expression in chondrocytes. In agreement with these findings, knockdown of *Tnxb* in cartilages of $F8^{-/-}$ mice resulted in accelerated HA-like lesions. It has been previously reported that TNXB blocks the interaction between TGF-β and its receptor in endothelial cells. Since a potent role of TGF-β in chondrocyte differentiation, we examined the effect of TNXB on TGF-β signaling, but did not observe significant alterations in this signaling in *Tnxb*-KD chondrocytes (*Figure 6—figure supplement 2, A-B*). The FNIII repeats domain of TNXB undergoes alternative splicing to interact with different ECM proteins and growth factors, and then affects ECM network formation and three-dimensional collagen matrix stiffness (*Valcourt et al., 2015*). Thus, future study would benefit from exploring the effect of the FNIII repeats domain in HA development to further clarify the specific mechanisms of TNXB.

The process of HA development is accompanied by the chondrocyte apoptosis which in part promotes the ECM degeneration (*Pulles et al., 2017*; *Hooiveld et al., 2003b*). We also observed remarkably increased apoptosis following joint bleeding in $F8^{-/-}$ mice. Interestingly, *Tnxb* knockdown could significantly induce apoptosis in chondrocytes *in vitro*, and even aggravate apoptosis in mouse HA cartilage, by enhancing expression of markers associated pro-apoptosis (Bax, C-Caspase3) and suppressing anti-apoptotic proteins (Bcl-2). In TNXB family, TNC was reported to inhibit human chondrosarcoma cell apoptosis by activation of AKT (*Jang and Chung, 2005*). PI3K signaling pathway regulated AKT phosphorylation to suppress apoptosis (*Tan et al., 2015*; *Testa and Bellacosa, 2001*). Our KEGG analysis of DMGs showed the potential association of PI3K/AKT signaling with HA. We then detected much lower phosphorylation level of AKT1 in *Tnxb*-KD chondrocytes and HA models. These results are in line with previous report showing that the expression of p-AKT1 was down-regulated in the articular cartilage of patients with HA compared with that in patients with OA (*Zheng et al., 2023*). Notably, we further demonstrated that AKT agonist effectively restored the abnormal Mmp13 expression and apoptosis induced by *Tnxb* knockdown. Recent studies have shown that AKT signaling affected cartilage metabolism by regulating Mmp13 expression (*El Mabrouk et al., 2007*; *Lu et al., 2021*). These findings indicate that AKT1 activation mediates the effect of TNXB on HA chondrocytes apoptosis and ECM catabolism. Thus, therapeutic effect of AKT agonist on HA development deserves further study.

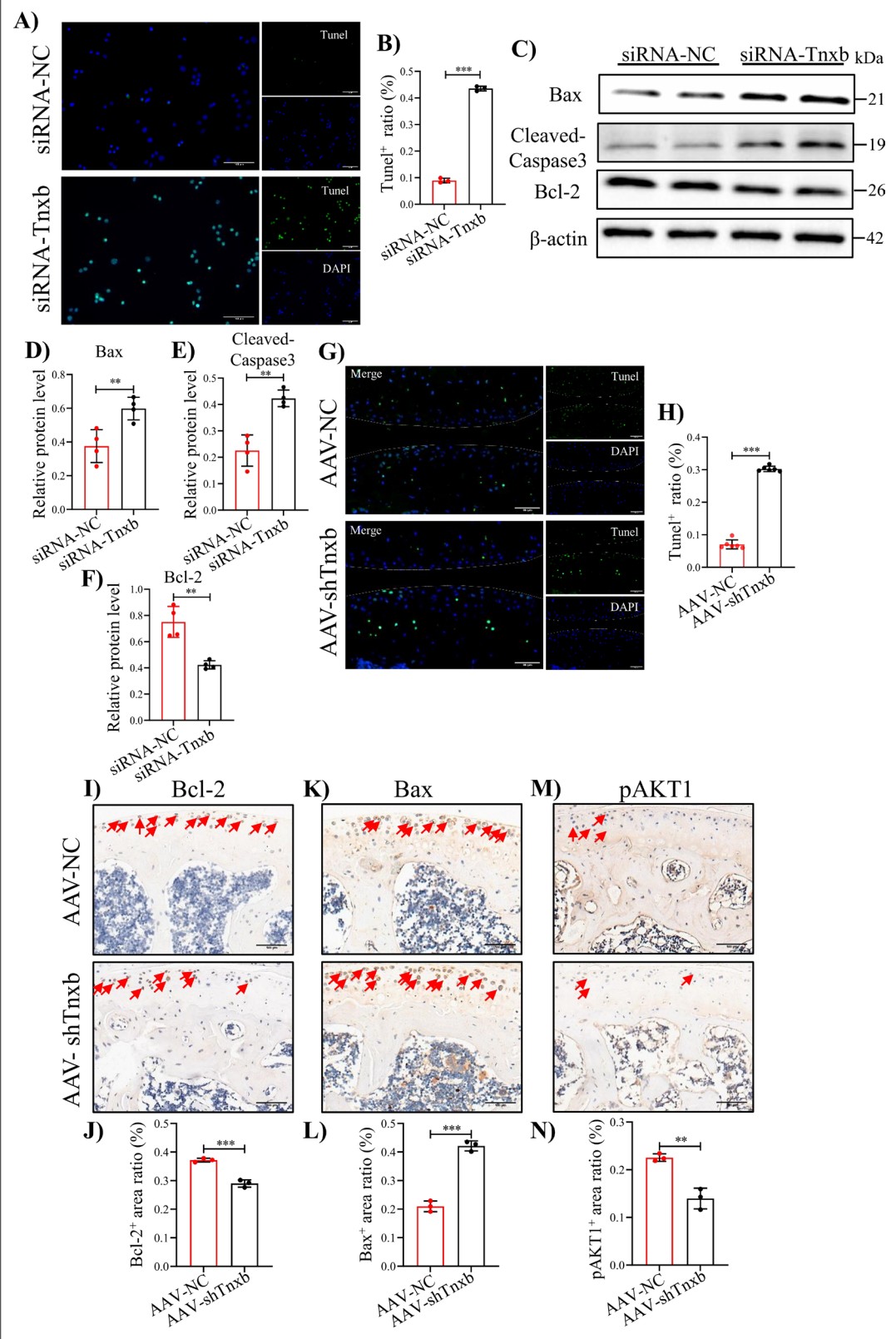

**Figure 5.** *Tnxb* knockdown increases apoptosis in chondrocytes and HA cartilages. (**A**) Representative images of Tunel staining in *Tnxb*-KD chondrocytes (**B**) Quantification for Tunel positive cells. (**C–F**) Western blot and corresponding quantification analysis for Bax, Cleaved-Caspase3, and Bcl-2. (**G**) Tunel staining for apoptosis in articular cartilage from *Tnxb*-KD HA mice. (**H**) Quantification for Tunel-positive cells in articular cartilage.

*Figure 5 continued on next page*

*Figure 5 continued*

Representative IHC staining of Bax (**I**), Bcl-2 (**K**), and p-AKT1 (**M**) in articular cartilage from *Tnxb*-KD HA mice. Corresponding quantification of the proportion of (**J**) Bax, (**L**) Bcl-2 and (**N**) p-AKT1-positive regions. Red arrows indicate positive cells. Scale bar: 100 µm. Data were presented as means ± SD; n ≥ 3 in each group. And analyzed by two-tailed unpaired parametric Student's t test, **$p < 0.01$, ***$p < 0.001$.

The online version of this article includes the following source data and figure supplement(s) for figure 5:

**Source data 1.** Original file for the western blot analysis of β-actin in *Figure 5C*.

**Source data 2.** Original file for the western blot analysis of Bax in *Figure 5C*.

**Source data 3.** Original file for the western blot analysis of Bcl-2 in *Figure 5C*.

**Source data 4.** Original file for the western blot analysis of Cleaved-Caspase 3 in *Figure 5C*.

**Source data 5.** PDF containing *Figure 5C* and original scans of the relevant western blot analysis (anti-β-actin, anti-Bax, anti-Bcl-2, and anti-Cleaved-Caspase 3) with highlighted bands and sample labels.

**Figure supplement 1.** Tunel staining for apoptosis in articular cartilage of *F8*[-/-] mice at (**A**) 4 and (**B**) 8 weeks following injury.

## Conclusions

Conclusively, our study provides the first comprehensive methylation profile in the progression of knee HA, and shows that abnormal expression of TNXB is important in HA cartilage degeneration by suppressing the activation of AKT1. These findings provide a new insight into mechanisms responsible for HA disease and suggest TNXB as a potentially novel therapeutic target for HA treatment.

## Materials and methods

### Cartilage specimen collection

Osteoarthritis (OA) is often used as 'disease' control to reveal the characteristics in HA (*Cooke et al., 2018*; *Kalebota et al., 2022*), although the mechanistic and phenotypic are different between HA and OA. In this study, 5 HA and 5 OA knee cartilage specimens were collected from patients undergoing total knee replacement surgery. Knee cartilage tissues were dissected and then rapidly frozen in liquid nitrogen or fixed in 4% paraformaldehyde. All subjects were Chinese Han and their demographic characteristics were listed in *Supplementary file 1*. The human cartilage samples were obtained from Department of Orthopedic Surgery at The First Affiliated Hospital of Zhejiang Chinese Medical University. This study was approved by the Ethics Committee of the First Affiliated Hospital of Zhejiang Chinese Medical University (2019-ZX-004–02).

### Magnetic resonance image (MRI)

MRI examination: a GE 3.0T Signa Excite superconducting MRI machine with a 32-channel body coil, a gradient field of 24 mT/m, and a creep rate of mT/(m.s) as well as a Siemens Verio 3.0T superconducting magnetic resonance imaging machine with a 16-channel phased array coil were applied. An ultrashort echo time (UTE) pulse sequence (echo time 0.07 ms) was performed in the sagittal plane of the knee joint of the affected limb. Scanning parameters: layer thickness of 0.8 mm, field of view of 320 mm×320 mm, matrix of 160 mm×160 mm, intra-layer resolution of 0.6 mm×0.6 mm, and excitation of 2 times. Scanning software: NooPhase Wrap, Variable Bandwidth, Tailored RF.

### Genome-wide DNA methylation profiling

Genomic DNA was prepared from cartilage specimens using QIAamp DNA Blood Mini Kit (QIAGEN, Germany) according to the manufacturer's instructions and was stored at –80 °C until used. Then bisulfite treatment of genomic DNA was performed with the EZ DNA methylation kit (Zymo Research) according to manufacturer's procedure. Genome-wide DNA methylation patterns were evaluated by Infinium Human Methylation 850 K BeadChips (Illumina), which determine the methylation levels of 853,307 CpG sites. R/Bioconductor (version 3.3.3) package ChAMP was used to process the Illumina intensity data (IDAT) files from the chip. DNA methylation level of each CpG site was described as a β value, ranging from zero (representing fully unmethylated) to one (representing fully methylated). Probes that had a detection *P* value of>0.01 and those located on the X and Y chromosome were

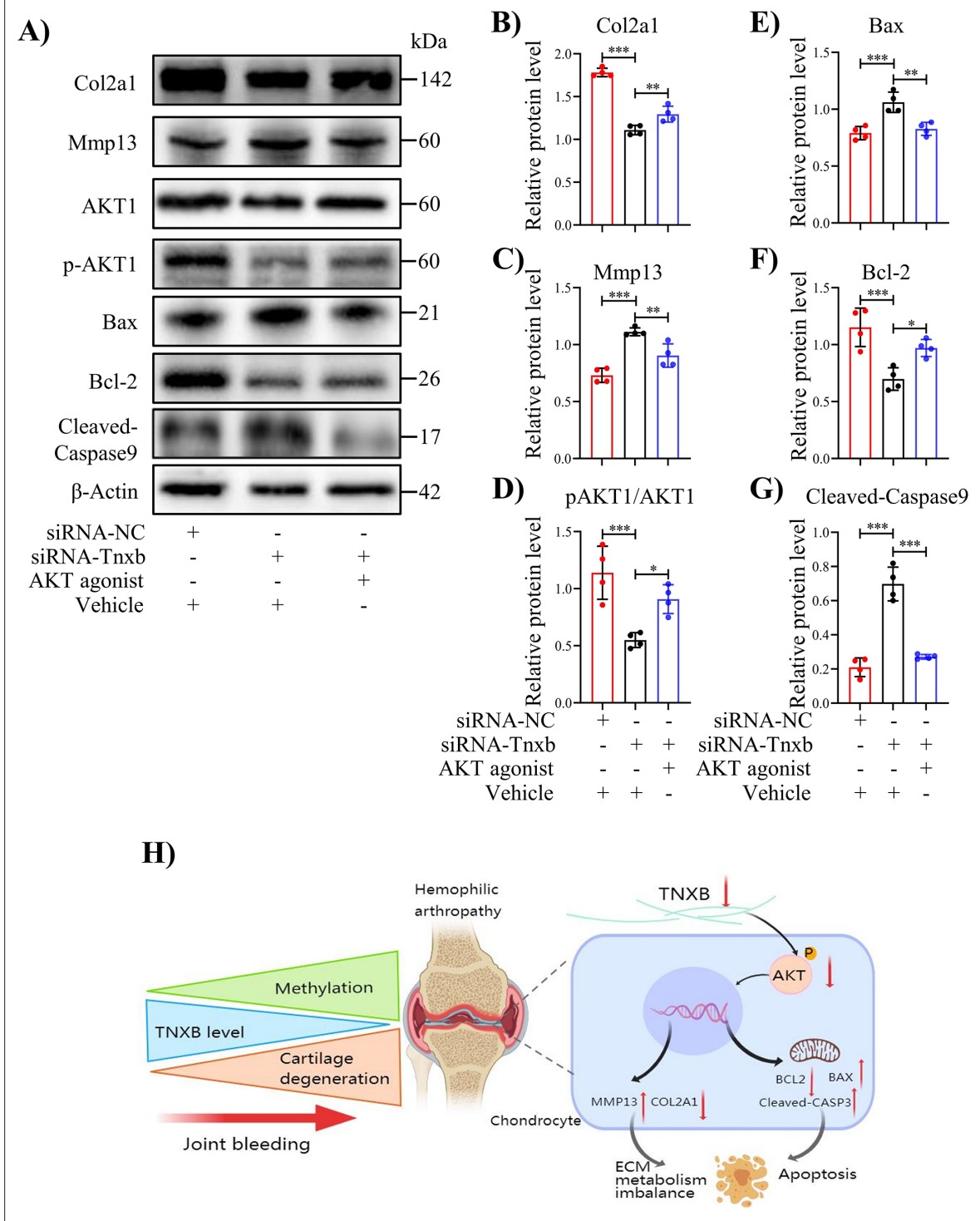

**Figure 6.** Treatment with AKT agonist decreases apoptosis in *Tnxb*-KD chondrocytes. (**A**) Western blot analysis for Col2a1, Mmp13, Bax, Bcl-2, p-AKT1, AKT1 and Cleaved-Caspase9 in *Tnxb*-KD chondrocytes treated with SC79 (0 or 10 μM) for 24 h. (**B–G**) Corresponding quantification of these proteins. (**H**) Schematic of the role of TNXB in the pathogenesis of HA cartilage degeneration. Data were presented as means ± SD; n = 4 per group. And analyzed by two-tailed unpaired parametric Student's t test or one-way ANOVA with Tukey's multiple comparisons test, $*p < 0.05$, $**p < 0.01$, $***p < 0.001$.

The online version of this article includes the following source data and figure supplement(s) for figure 6:

**Source data 1.** Original file for the western blot analysis of β-actin in *Figure 6A*.

**Source data 2.** Original file for the western blot analysis of Col2a1 in *Figure 6A*.

*Figure 6 continued on next page*

*Figure 6 continued*

**Source data 3.** Original file for the western blot analysis of Mmp13 in *Figure 6A*.

**Source data 4.** Original file for the western blot analysis of AKT1 in *Figure 6A*.

**Source data 5.** Original file for the western blot analysis of pAKT1 in *Figure 6A*.

**Source data 6.** Original file for the western blot analysis of Bax in *Figure 6A*.

**Source data 7.** Original file for the western blot analysis of Bcl-2 in *Figure 6A*.

**Source data 8.** Original file for the western blot analysis of Cleaved-Caspase9 in *Figure 6A*.

**Source data 9.** Original file for the western blot analysis of Cleaved-Caspase9-β-actin in *Figure 6A*.

**Source data 10.** PDF containing *Figure 6A* and original scans of the relevant western blot analysis (anti-β-actin, anti-Col2a1, anti-Mmp13, anti-AKT1, anti-pAKT1, anti-Bax, anti-Bcl-2, anti-Cleaved-Caspase 9, and anti-Cleaved-Caspase 9-β-actin) with highlighted bands and sample labels.

**Figure supplement 1.** Western blot examined the function of AKT1 and p-AKT1 agonist SC79.

**Figure supplement 1—source data 1.** Original file for the western blot analysis of β-actin in *Figure 6—figure supplement 1*.

**Figure supplement 1—source data 2.** Original file for the western blot analysis of AKT1 in *Figure 6—figure supplement 1*.

**Figure supplement 1—source data 3.** Original file for the western blot analysis of pAKT1 in *Figure 6—figure supplement 1*.

**Figure supplement 1—source data 4.** PDF containing *Figure 6—figure supplement 1* and original scans of the relevant western blot analysis (anti-β-actin, anti-AKT1, and anti-pAKT1) with highlighted bands and sample labels.

**Figure supplement 2.** Western blot examined the expression of p-Smad2 in mouse chondrocytes treated with siRNA-NC or siRNA-*Tnxb*.

**Figure supplement 2—source data 1.** Original file for the western blot analysis of β-actin in *Figure 6—figure supplement 2*.

**Figure supplement 2—source data 2.** Original file for the western blot analysis of pSmad2 in *Figure 6—figure supplement 2*.

---

filtered away. We also removed SNP-related probes and all multi-hit probes. BMIQ (Beta MIxture Quantile dilation) algorithm was used to correct for the Infinium type I and type II probe bias. Furthermore, the regression models were adjusted for age and sex. Differentially methylated probes (DMPs) at significance of $p< 0.05$ after the Benjamini & Hochberg correction for multiple testing. DMRs were identified by Bumphunter with default settings. Additionally, we annotated the DMR-related genes (DMGs), and then performed Gene Ontology (GO) and Kyoto Encyclopedia of Genes (KEGG) pathway analysis by using the online tool DAVID (http://david.abcc.ncifcrf.gov/). The significant pathway/ GO term was identified with p-value <0.05.

## Mouse

The FVIII target gene knockout hemophilia ($F8^{-/-}$) mouse were purchased from the Shanghai Laboratory Animal Center of the Chinese Academy of Sciences. All mouse were kept in the Laboratory Animal Center of Zhejiang University of Chinese Medicine. The animals were placed in the controlled environmental condition with relatively constant air humidity and temperature as well as manual control of day and night replacement for 12 hr. The experiment was approved by the Experimental Animal Ethics Committee of Zhejiang University of Traditional Chinese Medicine (LZ12H27001).

## The mouse experimental model of HA

The HA model is constructed according to the description of *Hakobyan et al., 2008*. $F8^{-/-}$ male mouse were induced with 3% isoflurane, and 1% isoflurane was maintained under anesthesia. The right knee joint capsule of mouse was pierced with a 30 g needle for bleeding modeling, and the left knee of each animal was used as a control group. Mouse were sacrificed at 4 and 8 weeks after injury, and knee joints were collected for further experiments.

## Intra-articular injection

$F8^{-/-}$ male mouse were induced with 3% isoflurane, and 1% isoflurane was maintained under anesthesia. The right knee joint capsule of mouse was pierced with a 30 g needle for bleeding modeling. The needle of the insulin syringe was sagittally inserted into the intercondylar area of the mouse right knee joint, and one dose of 10 µL AAV-shTnxb (GenePharma, concentration: 1×10^10^/10 µL) or AAV-NC was injected. After 4 weeks, mouse was sacrificed, respectively, and the right knee joints were collected for further experiments.

## Micro-computed tomography (micro-CT) analysis

Micro-computed tomography (micro-CT) (Skyscan 1176, Bruker µCT, Kontich, Gelgium) was used to analyze the knee joints. The area between the proximal tibia growth plate and the tibial plateau was chosen as the region of interest. The parameters collected form µCT were percent bone volume (BV/TV, %), Trabecular thickness (Tb.Th, mm), Trabecular number (Tb.N, 1 /mm) and Trabecular Spacing(Tb.Sp, mm).

## Histological analysis

The human cartilage samples were successively fixed in 4% paraformaldehyde for 5 days, decalcified with 14% EDTA solution for 3 months, and the mouse samples were successively fixed in 4% paraformaldehyde for 3 days, decalcified with 14% EDTA solution for 14 days. These samples subsequently embedded in paraffin. Then 3-µm-thick sections at the medial compartment of the joints were cut for Alcian blue hematoxylin/Orange G (ABH) or Toluidine Blue staining to analyze the gross cartilage structural changes. Histomorphometry analysis was performed through Osteomeasure software (Decatur, GA).

## Immunohistochemistry

The immunohistochemistry (IHC) was examined to observe the expressions of protein in cartilage. Briefly, the deparaffinized sections were soaked in 0.3% hydrogen peroxide to block the activity of endogenous peroxidase, then blocked with normal goat serum (diluted 1:20) for 20 min at room temperature. Subsequently, the primary antibodies were added and incubated overnight at 4 ° C. The next day, the sections were treated with secondary antibodies for 30 min and positive staining was visible by using diaminobenzidine solution (Invitrogen, MD, United States). Then counterstaining was performed with hematoxylin for 5 s. Anti-Col2a1 (Abcam, ab34712, 1:200), Anti-Mmp13 (Abcam, ab39012, 1:300), Anti-Tnxb (Proteintech, 13595–1-AP, 1:50), Anti-Bax (Huabio, ET1603-34, 1:200), Anti-Bcl-2 (Huabio, ET1702-53, 1:200), Anti-p-AKT1 (Abclonal, AP0140, 1:200), Anti-Dnmt1(Abcam, 1 : 200), Anti-Dnmt3a (Abcam, 1 : 200), Anti-Dnmt3b (Huabio, 1 : 200) were used in this study.

## Cell isolation and culture

Primary mouse chondrocytes were obtained from the femoral head of 2-week-old C57BL/6-J mouse purchased from the Experimental Animal Center of Zhejiang Chinese Medical University. Mouse were sacrificed and disinfected with 75% ethyl alcohol. Specimens were isolated and rinsed by Phosphate Buffer Saline (PBS) 3 times. Then the cartilage tissues were digested with 0.25% collagenase P at 37 °C for 4 hr. Chondrocytes were cultured in DMEM/F-12 medium containing 10% fetal bovine serum (FBS) and 1% streptomycin/penicillin in 5% $CO_2$ at 37 °C for further experiment.

## Tunel assay

To evaluate the apoptotic cells in the chondrocytes, we performed a Tunel assay according to the manufacturer's guideline (Beyotime, catalog C1088). Briefly, following deparaffinage and rehydration, sections were permeabilized with DNase-free Proteinase K (20 µg/mL) for 15 min at 37 °C. Subsequently, slides were treated with Tunel solution and incubated at 37 °C for 1 hr in a dark environment and counterstained with DAPI (1:1000; Solarbio) for 10 min. Finally, Tunel positive cells were detected by fluorescence microscope.

## Transfection of siRNA-*Tnxb*

Chondrocytes were transiently transfected with siRNA targeting *Tnxb* (GenePharma) or negative-control siRNA (scrambled; GenePharma) using X-tremeGENE siRNA transfection reagent (Sigma) following the manufacturer's instructions. All experiments were performed 72 hr after transfection, and the most effective single siRNA was used for further experiments. After transfection for 48 hr, cells were treated with various concentrations of SC79 (10 µM, Selleck) for 24 hr.

## Cell immunofluorescence

The cells were fixed with 4% paraformaldehyde and blocked with goat serum for 1 hr. Then, cells were incubated with Col2a1 antibody (1:200; Abcam) and Mmp13 antibody (1:100; Abcam) overnight. The

next day, the knee joints samples and Chondrocytes with goat anti-rabbit antibody (1:1000; Invitrogen) conjugated with Alexa Fluor 488 for 1 hr. Cells were stained with DAPI (1:1000; Solarbio).

## Western blot analysis

Total proteins were extracted by RIPA buffer, and concentration of protein was determined by bicinchoninic acid (BCA, Beyotime, Shanghai, China). Proteins were separated on 10% SDS-PAGE gels and transferred to PVDF membranes. The membranes were then blocked with 5% skim milk for 1 hr, and were incubated with primary antibodies (Col2a1, Abclonal, A1560, 1:1500), (Mmp13, Abclonal, A11755, 1:1000), (Tnxb, Abclonal, A2535, 1:1000), (p-AKT1, Abclonal, AP0140, 1:1000), (AKT1, Huabio, ET1609-47, 1:1000), (Bcl-2, Huabio, ET1702-53, 1:1000), (Bax, Huabio, ET1603-34, 1:1000), (active-pro caspase-3, Huabio, ET1608-64, 1:1000), (Cleaved-Caspase9, Abclonal, A22672, 1:1000), (pSmad2, CST, 8828, 1:1000) overnight at 4 °C. The membranes were washed three times, and then cultured with the secondary antibody for 1 hr. The immunoreactivity was detected with the ECL substrate (Thermo Fisher Scientific, United States) on an Image Quant LAS 4000 (EG, United States). The grey value was calculated by the software of Image J. *β-actin* was used as an internal control in all western blot analysis.

## Quantitative RT-PCR

TRIzol reagent (Invitrogen, United States) was used according to the manufacturer's protocol to extract total RNA from chondrocytes. The total RNA quantity and purity was evaluated by using NanoDrop 2000 (Thermo Fisher Scientific, United States). Reverse transcription was carried out with cDNA Synthesis Kit (Bmake, Beijing, China). The quantitative real-time-polymerase chain reaction (qPCR) was conducted with SYBR Premix Ex Taq II (Takara, Dalian, China), and performed on on a QuantStudio 7 Flex Real-Time PCR System. All of the PCR reactions were repeated three times for each gene. The data were analyzed by the $2-\Delta\Delta CT$ method using *Actb* as the internal control. ***Supplementary file 2*** showed the primer sequences of target genes used in the current study.

## Treatment of DNA methyltransferase inhibitors

To investigate whether DNA methylation affects the expression of TNXB, we treated primary mouse chondrocytes with RG108 or 5-Aza-dc (0, 1, 10, and 25 µM) (RG-108, Beyotime, SD-1137; 5-Aza-2'-deoxycytidine, Beyotime, SD-1047) for 24 hr. The cells were then collected for qPCR to detect the mRNA level of *Tnxb*.

## Statistics

All experimental data of this subject were analyzed by GraphPad Prism software version 8.0, and the statistical results of measurement data were expressed as mean ± standard deviation. Two-tailed unpaired parametric Student's t test or nonparametric Mann-Whitney U test or one-way ANOVA with Tukey's multiple comparisons test was used for comparison between different groups. The difference was statistically significant with $p < 0.05$.

## List of abbreviations

HA: hemophilic arthropathy; DMRs: differentially methylated regions; DMGs: DMR-related genes; TNXB: Tenascin XB; DNMT: DNA methyl transferase; ABH: Alcian blue hematoxylin/Orange G; ECM: extracellular matrix; AAV: adeno-associated virus.

## Study approval

In the current study, the human cartilage samples were obtained from Department of Orthopedic Surgery at The First Affiliated Hospital of Zhejiang Chinese Medical University. This study was approved by the Ethics Committee of the First Affiliated Hospital of Zhejiang Chinese Medical University (2019-ZX-004–02). And all the animal operating procedures in this study were approved by the Experimental Animal Ethics Committee of Zhejiang Chinese Medical University (LZ12H27001).

## Acknowledgements

We appreciate the great help from the Public Platform of Medical Research Center, Academy of Chinese Medical Science, Zhejiang Chinese Medical University. This research has been partially supported by the National Natural Sciences Foundation of China (Grant nos. 82274280, 82104891, 82074457), Zhejiang Provincial Natural Science Foundation of China (LQ22H270006, LR23H270001).

## Additional information

### Competing interests

Di Chen: Reviewing editor, eLife. The other authors declare that no competing interests exist.

### Funding

| Funder | Grant reference number | Author |
|---|---|---|
| National Natural Science Foundation of China | 82274280 | Hongting Jin |
| National Natural Science Foundation of China | 82074457 | Weidong Wang |
| National Natural Science Foundation of China | 82104891 | Yuliang Huang |
| Natural Science Foundation of Zhejiang Province | LQ22H270006 | Jiali Chen |
| Natural Science Foundation of Zhejiang Province | LR23H270001 | Hongting Jin |

The funders had no role in study design, data collection and interpretation, or the decision to submit the work for publication.

### Author contributions

Jiali Chen, Conceptualization, Funding acquisition, Writing – original draft; Qinghe Zeng, Data curation, Writing – original draft; Xu Wang, Rui Xu, Data curation, Methodology; Weidong Wang, Funding acquisition, Visualization, Methodology; Yuliang Huang, Funding acquisition, Methodology; Qi Sun, Validation, Methodology; Wenhua Yuan, Investigation, Visualization; Pinger Wang, Visualization, Methodology; Di Chen, Peijian Tong, Writing – review and editing; Hongting Jin, Conceptualization, Funding acquisition, Writing – review and editing

### Author ORCIDs

Jiali Chen ⓘ https://orcid.org/0009-0009-9876-6103
Qinghe Zeng ⓘ http://orcid.org/0000-0001-7543-0329
Di Chen ⓘ http://orcid.org/0000-0002-4258-3457
Hongting Jin ⓘ http://orcid.org/0000-0001-8795-0874

### Ethics

In the current study, the human cartilage samples were obtained from Department of Orthoapedic Surgery at The First Affiliated Hospital of Zhejiang Chinese Medical University. This study was approved by the Ethics Committee of the First Affiliated Hospital of Zhejiang Chinese Medical University (2019-ZX-004-02).

And all the animal operating procedures in this study were approved by the Experimental Animal Ethics Committee of Zhejiang Chinese Medical University (LZ12H27001).

Reviewer #1 (Public review): https://doi.org/10.7554/eLife.93087.3.sa1
Reviewer #2 (Public review): https://doi.org/10.7554/eLife.93087.3.sa2
Author response https://doi.org/10.7554/eLife.93087.3.sa3

## Additional files

### Supplementary files
• Supplementary file 1. Basic characteristics of subjects.

• Supplementary file 2. Primer sequences for qPCR.

• Supplementary file 3. Identification and functional enrichment analysis of differentially methylated genes between HA and OA cartilages. (a) DMRs between HA and OA cartilages (b) GO pathway enrichment analysis of DMRs-related gene (Top 10 pathways) (c) KEGG pathway enrichment analysis of DMRs-related gene (Top 10 pathways).

• MDAR checklist

### Data availability
All data generated or analysed during this study are included in the manuscript and supporting files; source data files have been provided for Figures 1 to 6.

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
