## [Editor Report · eLife assessment]

This **important** study identifies the TNXB-AKT pathway as a potential mechanism underlying hemophilia-associated cartilage degeneration. The evidence supporting the conclusions is **convincing**, with murine and human patient evidence as well as genome-wide DNA methylation analysis. This paper would be of interest to cell biologists and biochemists working on the field of musculoskeletal disorders.

---

## [Referee Report · Reviewer #1 (Public review)]

Summary:

Chen and colleagues first compared the cartilage tissues collected from OA and HA patients using histology and immunostaining. Then, a genome-wide DNA methylation analysis was performed, which informed the changes of a novel gene, TNXB. IHC confirmed that TNXB has a lower expression level in HA cartilage than OA. Next, the authors demonstrated that TNXB levels were reduced in HA animal model, and intraarticular injection of AAV carrying TNXB siRNA induced cartilage degradation and promoted chondrocyte apoptosis. Based on KEGG enrichment, histopathological analysis, and western blot, the authors also showed the relationship between TNXB and AKT phosphorylation. Lastly, AKT agonist, specifically SC79 in this study, was shown to partially rescue the changes of in vitro-cultured chondrocytes induced by Tnxb knock-down. Overall, this is an interesting study and provided sufficient data to support their conclusion.

Strengths:

(1) Both human and mouse samples were examined.

(2) The HA model was used.

(3) genome-wide DNA methylation analysis was performed.

Weaknesses:

(1) In some experiments, the selection of the control groups was not ideal.

(2) More details on analyzing methods and information on replicates need to be included.

(3) Discussion can be improved by comparing findings to other relevant studies.

(4) The use of transgenic mice with conditional Tnxb depletion can further define the physiological roles of Tnxb.

---

## [Referee Report · Reviewer #2 (Public review)]

Summary:

This manuscript mainly studied the biological effect of tenascin XB (TNXB) on hemophilic arthropathy (HA) progression. Using bioinformatic and histopathological approaches, the authors identified the novel candidate gene TNXB for HA. Next, authors showed that TNXB knockdown lead to chondrocyte apoptosis, matrix degeneration and subchondral bone loss in vivo/vitro. Furthermore, AKT agonist promoted extracellular matrix synthesis and prevented apoptosis in TNXB knockdown chondrocytes.

Strengths:

In general, this study significantly advances our understanding of HA pathogenesis. The authors utilize comprehensive experimental strategies to demonstrate the role of TNXB in cartilage degeneration associated with HA. The results are clearly presented, and the conclusions appear appropriate.

Weaknesses:

Additional clarification is required regarding the gender of the F8-/- mouse in the study. Is the mouse male or female?

---

## [Author Response]

The following is the authors’ response to the original reviews.

**Public Reviews:**

**Reviewer #1 (Public Review):**
Summary:Chen and colleagues first compared the cartilage tissues collected from OA and HA patients using histology and immunostaining. Then, a genome-wide DNA methylation analysis was performed, which informed the changes of a novel gene, TNXB. IHC confirmed that TNXB has a lower expression level in HA cartilage than OA. Next, the authors demonstrated that TNXB levels were reduced in the HA animal model, and intraarticular injection of AAV carrying TNXB siRNA induced cartilage degradation and promoted chondrocyte apoptosis. Based on KEGG enrichment, histopathological analysis, and western blot, the authors also showed the relationship between TNXB and AKT phosphorylation. Lastly, AKT agonist, specifically SC79 in this study, was shown to partially rescue the changes of in vitro-cultured chondrocytes induced by Tnxb knock-down. Overall, this is an interesting study and provided sufficient data to support their conclusion.Strengths:(1) Both human and mouse samples were examined.(2) The HA model was used.(3) Genome-wide DNA methylation analysis was performed.Weaknesses:(1) In some experiments, the selection of the control groups was not ideal.

Thank you for comments. The reviewer raised the concerns about using human OA cartilage as control, instead of health cartilage. This is an important detail we didn’t describe in the previous version. We have added our explanation in revised Methods.

(2) More details on analyzing methods and information on replicates need to be included.

We greatly appreciate your careful review and helpful suggestions. We have added detailed information to our revised draft.

(3) Discussion can be improved by comparing findings to other relevant studies.

Thank the reviewer very much for the opportunity to improve our manuscript. We have improved discussions as reviewer suggested in Recommendation 13.

(4) The use of transgenic mice with conditional Tnxb depletion can further define the physiological roles of Tnxb.

Thanks for this valuable comment. We understand that conditional *Tnxb*-KO mice is much helpful for the study of biological roles of Tnxb, and it will be constructed and used in our future studies.

**Recommendations For the Authors:**
(1) Please add more information about HA such as incidence to highlight the importance of the study.

We greatly appreciate your careful review and helpful suggestions. We have provided more information about the importance of HA study in revised Introduction. Please see lines 68-71 and 81-90.

(2) Please justify the use of OA cartilage, instead of normal tissues, as the control.

Thanks for your suggestion. We certainly would have liked to use healthy cartilage as control, but we were extremely difficult to obtain enough control samples from healthy individuals. Despite the mechanistic and phenotypic differences between HA and OA, OA is often used as “disease” control to reveal the characteristics in HA^1,2^. Thus, we measured cartilage degeneration and DNA methylation difference in HA and OA patients. We have provided the statement and evidence in revised manuscript. Please see lines 327-328.

(3) Please provide details of how to calculate the Cartilage wear area ratio in Figure 1D, and measure the positive staining area in Figure 1F.

We apologize for the issue you pointed out. Here, we provide detailed information for how positively stained areas are calculated. Specifically, in Figure 1D, we obtained the cartilage area ratio by calculating the ratio of blue cartilage staining area to the whole tissue area by using ImageJ software. In Figure 1F, the area of positive staining was determined upon secondary antibody treatment and color development using DAB chromogen (brown stain). We then obtained the positive staining area ratio by calculating the ratio of positive staining area to the whole cartilage area by using ImageJ software.

(4) Please label the location of hemorrhagic ferruginous deposits in Figure 1.

Thank you for your valuable suggestion. We have used black arrows to indicate hemorrhagic ferruginous deposits in revised Figure 1A.

(5) Please define the meaning of "n" in all figure legends, such as technical or biological replicates.

Thanks for your suggestion. We have defined the meaning of "n" in all figure legends in revised manuscript.

(6) In Figure 3, please increase the font size of B, D, F, H, and J. The same applies to other figures.

Thank you for your valuable suggestion. We have increased the font size of figures in our revised manuscript.

(7) Line 327, "(Figure 1, F and G)" should be Figure 2F, G.

Thanks for your reminding. We have corrected it in the revision. Please see lines 163.

(8) Reduced TNXB levels in human HA cartilage are one of the major findings in this study. Currently, only semi-quatative IHC was used to draw the conclusion. A second method, such as real-time PCR or western blot, is required.

Thanks for your suggestion. We feel very sorry that we did not have enough samples of human HA cartilages for qPCR and WB experiments, due to severe erosion of the HA cartilage. We have pointed out this limitation in revised drafts. Please see lines 260-265.

(9) Figure 3 shows that reduced Tnxb was accompanied by the increased Dnmt1. In addition, this study is about methylation. Have the authors tested the change of Dnmt1 levels when Tnxb was knocked down?

Thanks for your suggestion. According to the reviewer's suggestion, we have tested the expression of Dnmt1 in *Tnxb*-KD chondrocytes, and no significant alteration was observed. Please see the following Figure.

**Author response image 1. sa3fig1:** Figure Legend: Representative IHC staining of Dnmt1 in articular cartilage from *Tnxb*-KD HA mice. Corresponding quantification of the proportion of Dnmt1 positive regions. Red arrows indicate positive cells. Scale bar: 100 μm. Data were presented as means ± SD; n = 5 in each group. ns = no significance by unpaired Student’s t test.

(10) Also, is there a causal relationship between Tnxb levels and the distribution of methylation levels? Any related study was performed?

Following the valuable suggestion of the reviewer, we used two well-known DNA methyltransferase inhibitors (RG108 or 5-Aza-dc)^3^ to examine whether DNA methylation regulates transcriptional expression of TNXB. We found that both inhibitors significantly up-regulated *Tnxb* mRNA level. We have added this result to the revised Figure 4-figure supplement 1 and draft (lines 185-190 and 476-480).

(11) In Figure 6, what was the control of "AKT agnost" group?

Thank you for your suggestion. We feel sorry for our negligence and we have added the vehicle group as a control for AKT agonists in Figure6 in our revised manuscript.

(12) Previous studies have reported the involvement of TNXB in TGF-β signaling. Have the authors examined the effect of TNXB on TGF-β signaling in chondrocytes?

Thank you for your suggestion. Here, we examined the expression of TGF-β signaling in Tnxb-KD chondrocyte and no significant changes were observed. We have discussed this result in revised draft (lines 290-294). We have added this result to the revised Figure 6-figure supplement 2.

(13) Discussion can be improved. For example, have previous studies reported the association between TNXB and methylation in other cells/tissues? In addition to apoptosis, are there other potential mechanisms underlying the protective role of TNXB in chondrocytes?

Thank you for your valuable comments. Previous studies have shown the different DNA methylation of TNXB in whole blood from rheumatoid arthritis patients and in retinal pigment epithelium from patients with age-related macular degeneration^4,5^. Herein, we were the first to report the association between DNA methylation of TNXB and HA cartilage degeneration. As for TNXB, there are limited public studies regarding physiological function of TNXB, among which mostly report the effect of TNXB on extracellular matrix organization^6,7^. In our work, we found that TNXB regulated the phosphorylation of AKT. Since previous reports showed AKT controlled the expression of Mmp13^8^, we thought that TNXB might regulated the chondrocyte extracellular matrix organization, in addition to its function on apoptosis. We have discussed these in revised manuscript (lines 277-279, and 310-313).

(14) The manuscript writing needs to be improved. Typos and grammar issues were noted.

Thanks. We have modified and polished our language and we hope the revised version could be acceptable for you.

**Reviewer #2 (Public Review):**
Summary:This manuscript mainly studied the biological effect of tenascin XB (TNXB) on hemophilic arthropathy (HA) progression. Using bioinformatic and histopathological approaches, the authors identified the novel candidate gene TNXB for HA. Next, the authors showed that TNXB knockdown leads to chondrocyte apoptosis, matrix degeneration, and subchondral bone loss in vivo/vitro. Furthermore, AKT agonists promoted extracellular matrix synthesis and prevented apoptosis in TNXB knockdown chondrocytes.Strengths:In general, this study significantly advances our understanding of HA pathogenesis. The authors utilize comprehensive experimental strategies to demonstrate the role of TNXB in cartilage degeneration associated with HA. The results are clearly presented, and the conclusions appear appropriate.Weaknesses:Additional clarification is required regarding the gender of the F8-/- mouse in the study. Is the mouse male or female?

We feel sorry that we did not provide enough information about the gender of the *F8^-/-^* mouse in the previous draft. Here, we used male *F8^-/-^* mice as the study subjects for our experiments. Hemophilia A is predominantly seen in males because of the X chromosome linkage^9^.

**Recommendations For The Authors:**
Some issues need to be addressed in the manuscript:(1) During the progression of HA, in addition to cartilage degeneration, synovial hypertrophy and inflammation are also significant symptoms. How is the expression of TNXB in HA synovium?

Thank you for your valuable comments. According to the reviewer's suggestion, we tested the expression of TNXB in the synovium, and there was no statistically significant difference in the expression level of TNXB in the synovium (Figure2-figure supplement 3) Please see lines 163-165.

(2) Lines 183-188. The methods of virus infection should be more detailed. What was the concentration of the AAVs injected? And how many doses were administrated?

Thank you for your suggestion. We have added an explanation of virus infection and injected doses in revised methods section (lines 390).

(3) Line 197-198. Could the author double-check the decalcification time for human cartilage samples? Is it for 3 months? Or for 3 weeks?

Thank you for your suggestion. We have reconfirmed the decalcification of human cartilage samples for 3 months.

(4) Line 343-344 "Above results suggest that TNXB might be protective against HA and its cartilage suppression is closely related to HA development." The conclusion is inappropriate, please revise it.

Thanks for your suggestion. We have revised this conclusion into “Above results suggest that the suppression of TNXB in cartilage is associated with HA development”. Please see lines 181-182.

(5) Line 326-327, the IHC staining for human samples is shown in Figure 2, not Figure 1. Please double check and revise it.

Thanks for your reminding. We feel sorry for our negligence and we have corrected it in the revision. Please see lines 163.

(6) For Figure 1B, it shows the MRI images of knee joints. However, the method section lacks details regarding the MRI imaging scan and analysis. Could the author include this information in the method section?

Thank you for your valuable comments. We have added the method of MRI imaging scan and analysis in revised Methods. Please see lines 337-346.

(7) In Figure 5, The statistical result of Bcl-2 is inconsistent with its Western blot band. Please check.

Thanks for your reminding. We have modified it in the revision.

(8) Please read through the text carefully to check for language problems. For example, in Line 68 "Our" not "our".

Thanks for your reminding. In revision, we have corrected it.

**Reviewer #3 (Public Review):**
Summary:The manuscript by Dr. Chen et al. investigates the genes that are differentially methylated and associated with cartilage degeneration in hemophilia patients. The study demonstrates the functional mechanisms of the TNXB gene in chondrocytes and F8-/- mice. The authors first showed significant DNA methylation differences between hemophilic arthritis (HA) and osteoarthritis through genome-wide DNA methylation analysis. Subsequently, they showed a decreased expression of the differentially methylated TNXB gene in cartilage from HA patients and mice. By knocking down TNXB in vivo and in vitro, the results indicated that TNXB regulates extracellular matrix homeostasis and apoptosis by modulating p-AKT. The findings are novel and interesting, and the study presents valuable information in blood-induced arthritis research.Strengths:The authors adopted a comprehensive approach by combining genome-wide DNA methylation analysis, in vivo and in vitro experiments using human and mouse samples to illustrate the molecular mechanisms involved in HA progression, which is crucial for developing targeted therapeutic strategies. The study identifies Tenascin XB (TNXB) as a central mediator in cartilage matrix degradation. It provides mechanistic insights into how TNXB influences cartilage matrix degradation by regulating the activation of AKT. It opens avenues for future research and potential therapeutic interventions using AKT agonists for cartilage protection in hemophilic arthropathy. The conclusions drawn from the study are clear and directly tied to the findings.Weaknesses:(1) The study utilizes a small sample size (N=5 for both osteoarthritis and hemophilic arthropathy). A larger sample size would enhance the generalizability and statistical power of the findings.

Thank you for pointing out this deficiency. Indeed, our sample size is relatively small, although the overall sample size was sufficient for statistical analyses. And we have added this limitation in discussion in revised manuscript. Please see line 260-263. Considering the small sample size, we subsequently performed functional validation study for TNXB, one of the most significant genes, and demonstrated that TNXB exerted critical impacts on chondrocytes apoptosis in HA pathogenesis *in vivo* and *in vitro*.

(2) The use of an animal model (*F8*^*-/-*^ mouse) to investigate the role of TNXB may not fully capture the complexity of human hemophilic arthropathy. Differences in the biology between species may affect the translatability of the findings to human patients.

Thank you for your valuable comments. We recognize that biological differences between species can affect the clinical translation of research findings. In our work, we sequenced human cartilage samples to obtain the differentially methylated gene-TNXB. Meanwhile, we demonstrated that protein expression of TNXB protein was significantly down-regulated in HA human cartilage and *F8^-/-^* transgenic mouse cartilage. The *F8^-/-^* transgenic mouse serves as a well-accepted model for the study of hemophilia, which is phenotypically similar to that of human patients suffering from the disease and spontaneously bleeds into the joints and soft tissues. Besides, this model mouse has been widely used in the study of hemophilia and hemophilic arthritis^9-11^.

(3) The study primarily focuses on TNXB as a central mediator, but it might overlook other potentially relevant factors contributing to cartilage degradation in hemophilic arthropathy. A more holistic exploration of genetic and molecular factors could provide a broader understanding of the condition.

Thanks for your suggestion. Since our human sample size is relatively small, we should interpret differentially methylated genes cautiously. Therefore, we mainly focused on the most top significant gene TNXB for functional study. In our further study, we will expand the sample size to more comprehensively explore the molecular mechanisms of HA.

**Recommendations For The Authors:**
The following are my suggestions:(1) Why do the authors choose to concentrate on the knee joint in the introduction when hemophilia, characterized by a deficiency in clotting factor F8, is recognized as a systemic disease?

Thank you for your valuable comments. Although hemophilia a systemic disease, approximately 80%-90% of bleeding episodes in patients with hemophilia occur within the musculoskeletal system, especially in the knee joint^12^.

(2) While Figure 1 illustrates distinct expressions of Dnmt1 and Dnmt3a, only Dnmt1 results are presented in HA mice models in Figure3. To address this, it is suggested that the expression of Dnmt3a be explored in animal models.

Thank you for your suggestion. According to the reviewer's suggestion, we examined the expression of Dnmt3a in mouse articular cartilage, and the expression level of Dnmt3a was significantly up-regulated in both the 4 weeks and 8 weeks model groups compared with the control group (Figure3). Please see line 180.

(3) In Figure 3, the sample size for Dnmt1 is smaller than the other indicators; therefore, supplementing the sample count is recommended.

Thanks for your reminding. We have corrected it in the revision.

(4) Regarding Figure 4G, a few apoptotic cells were observed in the AAV NC group. It is advised that this figure be reviewed for accuracy.

Thanks for your suggestion. In Figure 5D, the AAV-NC group is the case of needle-injected with AAV. Therefore, it is normal for apoptotic cells to appear in the cartilage layer.

(5) The authors concluded that TNXB plays a role in apoptosis and AKT signaling. Providing expression data for Caspase9 would be valuable to strengthen this assertion, as PI3K/AKT signaling directly influences its activation during apoptosis.

Thank you for your comments. We have examined the expression of Cleaved-Caspase9 protein, and found that knockdown of TNXB resulted in upregulation of Cleaved-Caspase9 protein expression, which was reversed by addition of SC79. This result has added in revised Figure 6 and manuscript. Please see line 229.

(6) Quantitative analysis of the differences between the two groups in Supplemental Figures is necessary.

Thank you for your suggestion. We have added the quantitative analysis of the differences between the two groups in Figure Supplements.

(7) With three major isoforms (homologs) of AKT in mammals-AKT1, 2, and 3 - why did the authors specifically focus on AKT1?

Thank you for your comments. Based on the results of the KEGG enrichment analysis of differential methylated genes, we investigated the role of PI3K/AKT pathway in apoptosis of HA chondrocytes. AKT is universally acknowledged as a core factor in the PI3K/AKT pathway that plays critical roles in various cellular activities such as cell proliferation, cell differentiation, cell apoptosis, metabolism and so on^13,14^, More notably, several studies demonstrated that in AKT family, Akt1 primarily was involved in regulation of chondrocyte survival and proteoglycan synthesis^15^. Therefore, we detected phosphorylation of AKT1 in HA cartilages and *TNXB*-KD chondrocytes, and found that TNXB regulation chondrocytes ECM and apoptosis by AKT1.

Reference:

(1) Cooke, E.J., Zhou, J.Y., Wyseure, T., Joshi, S., Bhat, V., Durden, D.L., Mosnier, L.O., and von Drygalski, A. (2018). Vascular Permeability and Remodelling Coincide with Inflammatory and Reparative Processes after Joint Bleeding in Factor VIII-Deficient Mice. Thromb Haemost 118, 1036-1047. 10.1055/s-0038-1641755.

(2) Kleiboer, B., Layer, M.A., Cafuir, L.A., Cuker, A., Escobar, M., Eyster, M.E., Kraut, E., Leavitt, A.D., Lentz, S.R., Quon, D., et al. (2022). Postoperative bleeding complications in patients with hemophilia undergoing major orthopedic surgery: A prospective multicenter observational study. J Thromb Haemost 20, 857-865. 10.1111/jth.15654.

(3) Weiland, T., Weiller, M., Kunstle, G., and Wendel, A. (2009). Sensitization by 5-azacytidine toward death receptor-induced hepatic apoptosis. J Pharmacol Exp Ther 328, 107-115. 10.1124/jpet.108.143560.

(4) Anaparti, V., Agarwal, P., Smolik, I., Mookherjee, N., and El-Gabalawy, H. (2020). Whole Blood Targeted Bisulfite Sequencing and Differential Methylation in the C6ORF10 Gene of Patients with Rheumatoid Arthritis. J Rheumatol 47, 1614-1623. 10.3899/jrheum.190376.

(5) Porter, L.F., Saptarshi, N., Fang, Y., Rathi, S., den Hollander, A.I., de Jong, E.K., Clark, S.J., Bishop, P.N., Olsen, T.W., Liloglou, T., et al. (2019). Whole-genome methylation profiling of the retinal pigment epithelium of individuals with age-related macular degeneration reveals differential methylation of the SKI, GTF2H4, and TNXB genes. Clin Epigenetics 11, 6. 10.1186/s13148-019-0608-2.

(6) Mao, J.R., Taylor, G., Dean, W.B., Wagner, D.R., Afzal, V., Lotz, J.C., Rubin, E.M., and Bristow, J. (2002). Tenascin-X deficiency mimics Ehlers-Danlos syndrome in mice through alteration of collagen deposition. Nat Genet 30, 421-425. 10.1038/ng850.

(7) Zhang, K., Wang, X., Zeng, L.T., Yang, X., Cheng, X.F., Tian, H.J., Chen, C., Sun, X.J., Zhao, C.Q., Ma, H., and Zhao, J. (2023). Circular RNA PDK1 targets miR-4731-5p to enhance TNXB expression in ligamentum flavum hypertrophy. FASEB J 37, e22877. 10.1096/fj.202200022RR.

(8) Guo, H., Yin, W., Zou, Z., Zhang, C., Sun, M., Min, L., Yang, L., and Kong, L. (2021). Quercitrin alleviates cartilage extracellular matrix degradation and delays ACLT rat osteoarthritis development: An in vivo and in vitro study. J Adv Res 28, 255-267. 10.1016/j.jare.2020.06.020.

(9) Weitzmann, M.N., Roser-Page, S., Vikulina, T., Weiss, D., Hao, L., Baldwin, W.H., Yu, K., Del Mazo Arbona, N., McGee-Lawrence, M.E., Meeks, S.L., and Kempton, C.L. (2019). Reduced bone formation in males and increased bone resorption in females drive bone loss in hemophilia A mice. Blood Adv 3, 288-300. 10.1182/bloodadvances.2018027557.

(10) Haxaire, C., Hakobyan, N., Pannellini, T., Carballo, C., McIlwain, D., Mak, T.W., Rodeo, S., Acharya, S., Li, D., Szymonifka, J., et al. (2018). Blood-induced bone loss in murine hemophilic arthropathy is prevented by blocking the iRhom2/ADAM17/TNF-alpha pathway. Blood 132, 1064-1074. 10.1182/blood-2017-12-820571.

(11) Vols, K.K., Kjelgaard-Hansen, M., Ley, C.D., Hansen, A.K., and Petersen, M. (2019). Bleed volume of experimental knee haemarthrosis correlates with the subsequent degree of haemophilic arthropathy. Haemophilia 25, 324-333. 10.1111/hae.13672.

(12) Lobet, S., Peerlinck, K., Hermans, C., Van Damme, A., Staes, F., and Deschamps, K. (2020). Acquired multi-segment foot kinematics in haemophilic children, adolescents and young adults with or without haemophilic ankle arthropathy. Haemophilia 26, 701-710. 10.1111/hae.14076.

(13) Garcia, D., and Shaw, R.J. (2017). AMPK: Mechanisms of Cellular Energy Sensing and Restoration of Metabolic Balance. Mol Cell 66, 789-800. 10.1016/j.molcel.2017.05.032.

(14) Johnson, J., Chow, Z., Lee, E., Weiss, H.L., Evers, B.M., and Rychahou, P. (2021). Role of AMPK and Akt in triple negative breast cancer lung colonization. Neoplasia 23, 429-438. 10.1016/j.neo.2021.03.005.

(15) Rao, Z., Wang, S., and Wang, J. (2017). Peroxiredoxin 4 inhibits IL-1beta-induced chondrocyte apoptosis via PI3K/AKT signaling. Biomed Pharmacother 90, 414-420. 10.1016/j.biopha.2017.03.075.